# ECHO: Efficient Coarse-Grained Hybrid Optimization — Clip at Batch, Learn at Token

## Abstract

Reinforcement learning (RL) for large language models (LLMs) typically employs token-level clipping of importance sampling ratios to ensure training stability. While effective at preventing catastrophic policy shifts, such fine-grained clipping often excessively truncates learning signals, limiting optimization efficiency. To address this limitation, we propose ECHO, a novel RL method that combines batch-level clipping with token-level importance sampling. Specifically, ECHO computes an average importance sampling ratio across the entire batch and uses this single clipping bound to modulate the gradient of each token. This batch-level approach preserves richer global reward information while retaining fine-grained token attribution, enabling gradients to capture more holistic reward structures and improve sample efficiency, leading to faster convergence and more stable training. Our method also provides a new perspective on how to define importance sampling ratios and reward shaping in RL for LLMs. Experimental results on both in-domain Math and reasoning benchmarks demonstrate that ECHO not only accelerates convergence but also achieves highly competitive performance, highlighting its efficiency and robustness for large-scale LLM alignment.

## 1 Introduction

Recent Large Language Models (LLMs), such as O1 (OpenAI-Team, 2024), R1 (DeepSeek-AI, 2025), and Kimi-k1.5 (Kimi Team, 2025), have achieved remarkable progress on complex reasoning tasks through reinforcement learning (RL). Training on large-scale tasks with verifiable rewards, including mathematical reasoning and programming, has been shown to significantly enhance the reasoning capabilities of these models (Shao et al., 2024; Hendrycks et al., 2021; He et al., 2024).Despite these advances, scaling RL for LLM training remains challenging, primarily due to the instability of optimization.

This instability arises from the distribution mismatch between new and old policies. Existing approaches typically adopt importance sampling clipping to alleviate this issue (Schulman et al., 2017; Shao et al., 2024), yet most operate at the token level (Zheng et al., 2025; Schulman et al., 2017; Shao et al., 2024). Token-level ratios often fail to capture the overall discrepancy between policies, and clipping at this granularity tends to suppress low-probability tokens. As a result, many valuable learning signals are discarded, thereby limiting optimization efficiency and slowing convergence (Wang et al., 2025; Shao et al., 2025).

As illustrated in Figure 1a, token-level clipping frequently results in an excessive proportion of tokens being truncated, even in cases where the overall sequence-level update remains relatively small (Wang et al., 2025; Shao et al., 2025). Such over-clipping diminishes the effective learning signal and constrains policy exploration, thereby slowing convergence and limiting achievable performance. Motivated by this limitation, prior work such as GSPO has proposed computing the importance sampling ratio at the sequence level, applying it consistently to both clipping and advantage weighting (Zheng et al., 2025). This coarse-grained approach effectively alleviates excessive clipping and yields a more stable optimization trajectory, suggesting that sequence-level ratios provide a more reliable measure of the distributional discrepancy between the new and old policies. Building upon this insight, and recognizing that modern PPO-style algorithms typically update policies on mini-batches, we introduce a batch-level clipping strategy, in which the importance sampling ratio is computed at the batch level and employed for clipping. This design mitigates instability caused

by large distributional gaps between policies, thereby improving both optimization efficiency and convergence behavior.

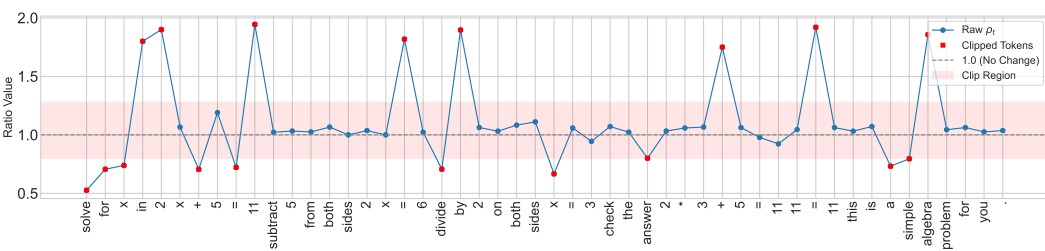

(a) Importance Ratios Per Token

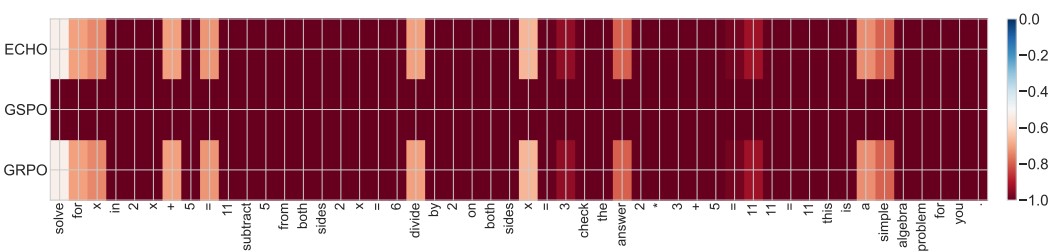

(b) Loss Terms by Algorithm

Figure 1: Comparison of GRPO, GSPO, and ECHO. (a) Clipping ranges for high- and low-ratio tokens, showing that GRPO clips many tokens while GSPO and ECHO avoid excessive clipping through coarser granularity. (b) Loss allocation across tokens, highlighting that GSPO applies uniform sequence-level ratios whereas ECHO distributes coarse-grained importance ratios more precisely, yielding a balanced and effective loss.

In addition to improving training stability, it is essential to preserve token-level im po a r tan ce. As illustrated in Figure 1b, methods such as GSPO assign the same sequence-level ratio to all tokens (Zheng et al., 2025), leading to identical weighted advantages across the sequence. This uniform treatment overlooks the fine-grained differences among tokens, which are often crucial for effective optimization. Capturing these token-level distinctions is therefore essential for retaining rich learning signals and enabling more precise credit assignment during training.

In this work, we introduce **ECHO**, a novel optimization method to address the instability of reinforcement learning for Large Language Models. ECHO consists of two key components: **Batch-level CLIP**, which computes importance sampling ratios at the level of PPO mini-batches and applies clipping at the batch scale to control distributional discrepancies and enhance training stability; and **Token-wise sampling**, which preserves fine-grained token-level importance, avoiding uniform weighting across sequences and ensuring that each token meaningfully contributes to optimization. Additionally, ECHO leverages batch-level ratios to reweight token-level advantages, smoothing the advantage distribution while maintaining relative differences across tokens, thereby improving both convergence speed and optimization efficiency. On benchmarks for mathematical and logical reasoning, ECHO achieves efficient convergence with fewer training steps, demonstrating its ability to enhance both training stability and model performance.

## 2 PRELIMINARIES

**Notation.** In this paper, we denote a Large Language Model (LLM) as a conditional policy $\pi_\theta$, parameterized by $\theta$. For an input query $q \in \mathcal{Q}$ sampled from a dataset $D$, the policy generates an output sequence $o = (o_1, \ldots, o_{|o|})$, where $|o|$ is the sequence length. Each token $o_t$ is regarded as an action, and the entire sequence $o$ corresponds to a trajectory. The probability of generating $o$ under

policy $\pi_\theta$ is:

$$\pi_\theta(o \mid q) = \prod_{t=1}^{|o|} \pi_\theta(o_t \mid q, o_{<t}).$$

Each $(q, o)$ pair receives a scalar reward $r(q, o) \in [0, 1]$ from a reward model $r_\phi$. For importance sampling (IS), we use the ratio:

$$\rho_t(\theta) = \frac{\pi_\theta(o_t \mid q, o_{<t})}{\pi_{\theta_{\text{old}}}(o_t \mid q, o_{<t})},$$

where $\pi_{\theta_{\text{old}}}$ is the behavior policy. Advantage estimates $A_t$ represent the relative value of action $o_t$ compared to the baseline.

## 2.1 PROXIMAL POLICY OPTIMIZATION (PPO)

Proximal Policy Optimization (PPO) (Schulman et al., 2017) is one of the most widely used policy gradient algorithms for reinforcement learning, particularly due to its balance between performance and stability. Instead of allowing unconstrained policy updates that may lead to performance collapse, PPO introduces a clipped surrogate objective to stabilize training.

$$\mathcal{J}_{\text{PPO}}(\theta) = \mathbb{E}_{q \sim D, \, o \sim \pi_{\theta_{\text{old}}}(\cdot | q)} \left[ \frac{1}{|o|} \sum_{t=1}^{|o|} \min \left( \rho_t(\theta) A_t, \, \text{clip}(\rho_t(\theta), 1 - \epsilon, 1 + \epsilon) A_t \right) \right], \quad (1)$$

where $\rho_t(\theta) = \frac{\pi_\theta(o_t | q, o_{<t})}{\pi_{\theta_{\text{old}}}(o_t | q, o_{<t})}$ is the token-level importance ratio and $\epsilon$ is the clipping hyperparameter. The clipping operation constrains $\rho_t(\theta)$ within $[1 - \epsilon, 1 + \epsilon]$, preventing excessively large policy updates that could destabilize training, while still allowing sufficient exploration when the ratio remains within the clipping bounds. This simple yet effective mechanism makes PPO a strong and widely adopted baseline for reinforcement learning from human or synthetic feedback.

## 2.2 GROUP RELATIVE POLICY OPTIMIZATION (GRPO)

Group Relative Policy Optimization (GRPO) (Shao et al., 2024) equation 2 eliminates the need for a separate value function by assigning *group-relative advantages* based on reward comparisons among multiple responses sampled per query. The optimization objective is:

$$\mathcal{J}_{GRPO}(\theta) = \mathbb{E}_{q \sim D, \, \{o_i\}_{i=1}^G \sim \pi_{\theta_{\text{old}}}} \left[ \frac{1}{G} \sum_{i=1}^G \frac{1}{|o_i|} \sum_{t=1}^{|o_i|} \min \left( \rho_{i,t}(\theta) \hat{A}_{i,t}, \, \text{clip}(\rho_{i,t}(\theta), 1 - \epsilon, 1 + \epsilon) \hat{A}_{i,t} \right) \right],$$
(2)

where $\rho_{i,t}(\theta)$ is the importance ratio, and $\hat{A}_{i,t}$ is the group-relative advantage computed from reward comparisons within $\{o_i\}$. By leveraging relative rewards within a sampled group, GRPO avoids training an explicit value function while still providing informative, low-variance advantage estimates.

## 2.3 GROUP SEQUENCE POLICY OPTIMIZATION (GSPO)

GSPO (Zheng et al., 2025) equation 3 replaces token-level importance ratios used in GRPO with a *sequence-level* ratio and clipping, directly comparing the likelihood of the entire output under the new and old policies. This design provides a more stable optimization signal by avoiding the compounding effect of token-level ratios.

Formally, the GSPO objective is:

$$\mathcal{J}_{GSPO}(\theta) = \mathbb{E}_{q \sim D, \, \{o_i\}_{i=1}^G \sim \pi_{\theta_{\text{old}}}} \left[ \frac{1}{G} \sum_{i=1}^G \min \left( s_i(\theta) \hat{A}_i, \, \text{clip}(s_i(\theta), 1 - \epsilon, 1 + \epsilon) \hat{A}_i \right) \right], \quad (3)$$

where the GSPO sequence-level importance ratio is defined as

$$s_i(\theta) = \left( \frac{\pi_\theta(o_i \mid q)}{\pi_{\theta_{\text{old}}}(o_i \mid q)} \right)^{\frac{1}{|o_i|}}. \quad (4)$$

## 3 ALGORITHM

### 3.1 ECHO: EFFICIENT COARSE-GRAINED HYBRID OPTIMIZATION

We extend GRPO with *batch-level clipping combined with token-wise sampling*, leading to our ECHO objective. For a mini-batch of size $B$, let each output sequence be $o_i = (o_{i,1}, \ldots, o_{i,|o_i|})$. For each token in response $o_i$, the token-level importance ratio is

$$\rho_{i,t}(\theta) = \frac{\pi_\theta(o_{i,t} \mid q, o_{i,<t})}{\pi_{\theta_{\text{old}}}(o_{i,t} \mid q, o_{i,<t})}. \tag{5}$$

### 3.2 BATCH-LEVEL CLIPPING AND EFFECTIVE TOKEN RATIOS

To stabilize optimization, we further aggregate token-level ratios across the batch and apply clipping, producing the effective token-level ratio $\tilde{\rho}_{i,t}(\theta)$ as described in Eq. equation 10.

**Batch-level average log-ratio.** We compute the batch-level average log-ratio across all tokens in the batch:

$$R_{\text{batch}}(\theta) = \exp\left(\frac{1}{\sum_{i=1}^{B} |o_i|} \sum_{i=1}^{B} \sum_{t=1}^{|o_i|} \Big( \log \pi_\theta(o_{i,t} \mid q, o_{i,<t}) - \log \pi_{\theta_{\text{old}}}(o_{i,t} \mid q, o_{i,<t}) \Big)\right), \tag{6}$$

where $B$ is the batch size and $|o_i|$ denotes the length of sequence $i$.

**Clipped batch-level ratio.** To stabilize optimization, we clip the batch-level ratio:

$$\bar{R}_{\text{batch}}(\theta) = \text{clip}\Big(R_{\text{batch}}(\theta), 1 - \epsilon, 1 + \epsilon\Big), \tag{7}$$

where $\epsilon$ is the clipping threshold.

**Effective token-level ratio.** The effective token-level ratio combines token-level importance sampling with the batch-level clipping:

$$\tilde{\rho}_{i,t}(\theta) = \rho_{i,t}(\theta) \cdot \frac{\bar{R}_{\text{batch}}(\theta)}{R_{\text{batch}}(\theta)^\dagger} \tag{8}$$

$$= \frac{\pi_\theta(o_{i,t} \mid q, o_{i,<t})}{\pi_{\theta_{\text{old}}}(o_{i,t} \mid q, o_{i,<t})} \cdot \frac{\text{clip}\Big(R_{\text{batch}}(\theta), 1 - \epsilon, 1 + \epsilon\Big)}{R_{\text{batch}}(\theta)^\dagger} \tag{9}$$

$$= \frac{\pi_\theta(o_{i,t} \mid q, o_{i,<t})}{\pi_{\theta_{\text{old}}}(o_{i,t} \mid q, o_{i,<t})} \cdot \frac{\text{clip}\bigg( \exp \frac{1}{\sum_{i,t} 1} \sum_{i,t} \log \rho_{i,t}(\theta), 1 - \epsilon, 1 + \epsilon \bigg)}{\bigg( \exp \frac{1}{\sum_{i,t} 1} \sum_{i,t} \log \rho_{i,t}(\theta) \bigg)^\dagger}. \tag{10}$$

Finally, the ECHO optimization objective is

$$\mathcal{J}_{\text{ECHO}}(\theta) = \mathbb{E}_{q \sim D, \{o_i\}_{i=1}^{G} \sim \pi_{\theta_{\text{old}}}} \left[ \frac{1}{\sum_{i=1}^{G} |o_i|} \sum_{i=1}^{G} \sum_{t=1}^{|o_i|} \min\Big( \tilde{\rho}_{i,t}(\theta) \, \hat{A}_{i,t}, \, \text{clip}\big(\tilde{\rho}_{i,t}(\theta), 1-\epsilon, 1+\epsilon\big) \, \hat{A}_{i,t} \Big) \right], \tag{11}$$

where $\hat{A}_{i,t}$ is the group-relative advantage as in GRPO equation 2, $\bar{R}_{\text{batch}}(\theta)$ (batch-level clipping) is defined in Eq. equation 7, and $\tilde{\rho}_{i,t}(\theta)$ (ECHO effective ratio) is defined in Eq. equation 10.

### 3.3 ECHO GRADIENT DERIVATION

The ECHO optimization objective is

$$\mathcal{J}_{\text{ECHO}}(\theta) = \mathbb{E}_{q \sim D, \{o_i\}_{i=1}^{G} \sim \pi_{\theta_{\text{old}}}} \left[ \frac{1}{\sum_{i=1}^{G} |o_i|} \sum_{i=1}^{G} \sum_{t=1}^{|o_i|} \min\Big( \tilde{\rho}_{i,t}(\theta) \, \hat{A}_{i,t}, \, \text{clip}\big(\tilde{\rho}_{i,t}(\theta), 1-\epsilon, 1+\epsilon\big) \, \hat{A}_{i,t} \Big) \right], \tag{12}$$

where $\tilde{\rho}_{i,t}(\theta) = \rho_{i,t}(\theta) \frac{\bar{R}_{\text{batch}}(\theta)}{R_{\text{batch}}(\theta)^\dagger}$.

In the unclipped region, the objective factorizes as $\mathcal{J}_{\text{ECHO}}(\theta) = \mathbb{E}[C(\theta) \cdot L(\theta)]$, where

$$C(\theta) = \frac{\bar{R}_{\text{batch}}(\theta)}{R_{\text{batch}}(\theta)^\dagger}, \tag{13}$$

$$L(\theta) = \frac{1}{N} \sum_{i,t} \rho_{i,t}(\theta) \hat{A}_{i,t}. \tag{14}$$

Applying the product rule and log-derivative trick yields the gradient:

$$\nabla_\theta \mathcal{J}_{\text{ECHO}}(\theta) = \mathbb{E}\left[ C(\theta) \frac{1}{N} \sum_{i,t} \rho_{i,t}(\theta) \Big( \hat{A}_{i,t} + \tfrac{L(\theta)}{R_{\text{batch}}(\theta)} \Big) \nabla_\theta \log \pi_\theta(o_{i,t} \mid q, o_{i,<t}) \right], \tag{15}$$

where $C(\theta)$ and $L(\theta)$ are defined in Eq. equation 14.

Each token's learning signal is modulated by the batch-level correction $C(\theta)$ and weighted by $\rho_{i,t}(\hat{A}_{i,t} + L(\theta)/R_{\text{batch}}(\theta))$ in Eq. equation 15, which combines individual advantage-weighted importance ratios with a $\rho$-scaled batch baseline to provide smoother and more stable gradient updates. The detailed derivation is provided in Appendix A.2.

## 3.4 ECHO GRADIENT ANALYSIS: COARSE-TO-FINE CONTROL

The ECHO optimization objective is defined in Eq. equation 11, and its corresponding gradient in the unclipped region is:

$$\nabla_\theta \mathcal{J}_{\text{ECHO}}(\theta) = \mathbb{E}\left[ C(\theta) \frac{1}{N} \sum_{i,t} \rho_{i,t}(\theta) \Big( \hat{A}_{i,t} + \tfrac{L(\theta)}{R_{\text{batch}}(\theta)} \Big) \nabla_\theta \log \pi_\theta(o_{i,t} \mid q, o_{i,<t}) \right],$$

where $C(\theta)$ and $L(\theta)$ are defined in Eq. equation 14.

This gradient naturally decomposes into two interacting components, revealing a coarse-to-fine control structure:

**1. Batch-level CLIP (Coarse-Grained Control)**   The batch-level correction factor

$$C(\theta) = \frac{\bar{R}_{\text{batch}}(\theta)}{R_{\text{batch}}(\theta)^\dagger}$$

is equal to 1 under normal circumstances, since the numerator and denominator are identical. Separately, Batch-level CLIP is triggered only when the batch-level ratio $R_{\text{batch}}(\theta)$ becomes excessively large or small, reducing the overall gradient magnitude to prevent destabilizing updates.

In addition, the term $L(\theta)/R_{\text{batch}}(\theta)$ in the gradient acts as a batch-level baseline adjustment, representing the weighted average of token advantages. Adding this batch-average to each token's advantage stabilizes the overall gradient across the batch while preserving token-level differences.

**2. Token-wise sampling (Fine-Grained Control)**   The token-level term

$$\rho_{i,t}(\theta) \hat{A}_{i,t}$$

preserves fine-grained learning signals for each token. Each token's contribution is weighted by its individual importance ratio $\rho_{i,t}(\theta)$, ensuring that low-probability but informative tokens can still effectively influence policy updates without being overshadowed by the batch-level adjustment.

By design, ECHO implements a principled coarse-to-fine gradient control: batch-level corrections constrain the overall update magnitude, ensuring stability during training, while token-level importance sampling retain fine-grained credit assignment, allowing each token to influence updates according to its importance. This architecture supports robust and efficient policy optimization in LLM reinforcement learning, striking a balance between convergence reliability and precise exploration.

# 4 EXPERIMENTS

## 4.1 TRAINING SETUP

We conduct experiments on Qwen2.5-Math-7B as the main backbone model. For comparison, we also evaluate Qwen Math 7B Base and Qwen Math 7B Instruct. The training dataset is OpenR1-Math-46K[1], which consists of diverse mathematical reasoning problems. For each prompt, we sample a group size of 8 responses using vLLM (Kwon et al., 2023) rollouts with temperature $T = 1.0$ and top-$p = 1.0$.

The maximum sequence length is set to $4096$ tokens, with a prompt budget of $1024$ and a response budget of $3072$. Overlong responses are truncated with a buffer of $512$ tokens and penalty factor $1.0$ to prevent excessive truncation of informative context.

We train for 290 steps with a global batch size of 512 across 8 nodes (64 H20 GPUs in total). Each update uses mini-batches of 64 prompts with micro-batch size 1 per GPU, enabled by dynamic batch sizing to optimize GPU utilization. For both ECHO and DAPO, we set the dynamic sampling batch size to be three times the training batch size, following standard practice to ensure sufficient exploration during rollout.

For clipping, we maintain the same upper and lower thresholds across all experiments, ensuring stability and comparability in the optimization process.

## 4.2 EVALUATION SETUP

We evaluate on nine widely used benchmarks, covering both in-domain mathematical reasoning and out-of-domain (OOD) general reasoning tasks:

**Mathematics benchmarks:** AIME2024 (MAA, 2024), AIME2025 (MAA, 2025), AMC2023 (LI et al., 2024), Math500 (Hendrycks et al., 2021), OlympiadBench (He et al., 2024), and Minerva (Lewkowycz et al., 2022).

**OOD benchmarks:** ARC-C (Clark et al., 2018), GPQA (Rein et al., 2024), and MMLU-Pro (Wang et al., 2024).

For AIME2024, AIME2025, and AMC2023, we report results using avg@32, where accuracy is averaged over 32 sampled generations per problem to reduce variance from stochastic sampling. For all other benchmarks, we adopt the conventional pass@1 metric. We uniformly employ math_verify to extract and evaluate the final answers across all benchmarks.

## 4.3 BASELINES

We compare ECHO against two categories of baselines: standard RL-based alignment methods and RLVR-style approaches.

**Standard Post-training Methods.** We evaluate several mainstream post-training methods, using the OpenR1-Math-46K dataset with reasoning traces.

- **SFT:** supervised fine-tuning on the training set, serving as a non-RL baseline.
- **GRPO** (Shao et al., 2024): reinforcement learning with token-level importance ratios, where per-sequence normalization is applied to reduce variance.
- **GSPO** (Zheng et al., 2025): optimizing with sequence-level importance ratios, combined with clipping to reduce variance.
- **DAPO** (Yu et al., 2025): Decoupled Advantage Policy Optimization, which separates advantage estimation from policy updates, and employs dynamic sampling for improved stability.
- **ECHO:** our proposed Efficient Coarse-Grained Hybrid Optimization, combining batch-level ratio clipping with token-level credit assignment, trained with dynamic sampling under the same hyperparameter setup as DAPO.

---

[1]https://hf-mirror.com/datasets/Elliott/Openr1-Math-46k-8192

**Previous RLVR methods.** We additionally benchmark against recent RLVR-based approaches trained from Qwen2.5-Math-7B. Specifically, Simple-RL (Zeng et al., 2025) applies rule-based reward optimization with vanilla RL; Oat-Zero (Liu et al., 2025) modifies GRPO by removing the standard deviation term in the advantage and skipping token-level normalization in the policy loss; PRIME-Zero (Cui et al., 2025) introduces implicit process rewards by leveraging outcome supervision; and OpenReasoner-Zero (Hu et al., 2025) provides a recent open-source RLVR implementation.

| Model | AIME24/25 | AMC | MATH | Minerva | Olympiad | ARC-c | GPQA | MMLU-Pro |
|---|---|---|---|---|---|---|---|---|
| Qwen-Math-7B | 11.5/4.9 | 31.3 | 43.6 | 7.4 | 15.6 | 18.2 | 11.1 | 16.9 |
| Qwen-Math-Instruct | 12.5/10.2 | 48.5 | 80.4 | 32.7 | 41.0 | 70.3 | 24.7 | 34.1 |
| *Previous RLVR Methods* | | | | | | | | |
| PRIME-Zero | 17.0/12.8 | 54.0 | 81.4 | 39.0 | 40.3 | 73.3 | 18.2 | 32.7 |
| OpenReasoner-Zero | 16.5/15.0 | 52.1 | 82.4 | 33.1 | 47.1 | 66.2 | 29.8 | **58.7** |
| Oat-Zero | 33.4/11.9 | 61.2 | 78.0 | 34.6 | 43.4 | 70.1 | 23.7 | 41.7 |
| SimpleRL-Zero | 27.0/6.8 | 54.9 | 76.0 | 25.0 | 34.7 | 30.2 | 23.2 | 34.5 |
| *Standard Post-training Methods* | | | | | | | | |
| SFT | 22.2/**22.3** | 52.8 | 82.6 | **40.8** | 43.7 | 75.2 | 24.7 | 42.7 |
| GRPO | 24.8/18.1 | 60.2 | 84.0 | 38.2 | 47.8 | 82.7 | 39.9 | 49.6 |
| GSPO | –/– | – | 29.8 | 15.8 | 8.6 | 72.0 | 29.3 | 30.3 |
| DAPO | 29.9/18.8 | **63.2** | **86.6** | 40.4 | 48.6 | 82.1 | 38.9 | 51.8 |
| *Our Methods* | | | | | | | | |
| ECHO | **34.1**/16.0 | 62.9 | 84.6 | 40.1 | **49.6** | **85.2** | **42.4** | 51.6 |

Table 1: Performance comparison across six math benchmarks (AIME 2024AIME 2025, AMC 2023, MATH, Minerva, Olympiad) and three out-of-distribution (OOD) datasets (ARC-c, MMLU-Pro, GPQA). Bold numbers indicate the best score in each column, and underlined numbers indicate the second-best. GSPO results for AIME and AMC are unavailable (–) due to severe sequence length collapse. Methods compared include Qwen Math 7B Base and its instruct variant, as well as recent RL approaches: Oat-Zero, PRIME-Zero, and OpenReasoner-Zero.

## 5 EXPERIMENTS RESULTS

### 5.1 MAIN RESULTS

**Performance.** Table 1 presents results on six Math benchmarks and three OOD benchmarks (ARC-c, GPQA, and MMLU-Pro). Overall, ECHO achieves competitive or superior performance compared to all baselines. In particular, ECHO delivers strong gains on challenging datasets such as AIME2024, Olympiad, ARC-c, and GPQA, as shown in Figure 3, where it consistently establishes new state-of-the-art results. On AMC2023 and MATH, ECHO also shows solid improvements, remaining competitive with DAPO. These results highlight that ECHO not only retains strong in-distribution capabilities but also generalizes better to out-of-distribution tasks, achieving the best overall balance among existing approaches.

**Training Dynamics.** In addition to final performance, ECHO exhibits stable and efficient training dynamics, as illustrated in Figure 2. Across experiments, ECHO converges faster than GRPO and DAPO, while avoiding the length collapse issue observed in GSPO. Notably, both policy entropy and sequence length remain stable throughout training, indicating that the optimization process is neither overly aggressive nor prematurely collapsed. This stability allows ECHO to reach high-performance plateaus more rapidly, while maintaining robustness against noisy or outlier updates. Overall, the empirical evidence demonstrates that ECHO achieves both strong generalization and

reliable convergence, establishing a new performance frontier for reinforcement learning based post-training methods.

## 5.2 RESULTS ANALYSIS

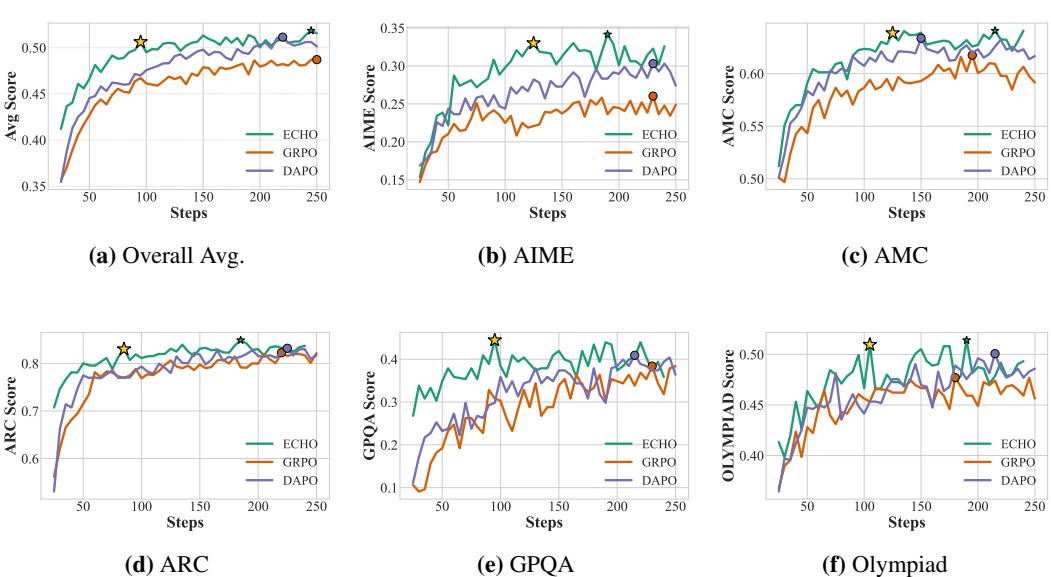

**(a)** Overall Avg.  **(b)** AIME  **(c)** AMC

**(d)** ARC  **(e)** GPQA  **(f)** Olympiad

Figure 2: (a) Line plots showing the overall average performance across nine benchmarks, including both Math and out-of-distribution (OOD) tasks. ECHO consistently achieves higher or comparable performance compared to GRPO and DAPO. Notably, ECHO reaches near-peak performance using fewer than half the training steps of DAPO, demonstrating more sample-efficient learning. This rapid convergence is evident across both in-domain Math datasets (e.g., ARC, AMC2023) and OOD benchmarks (e.g., GPQA, Olympiad), highlighting ECHO's ability to quickly adapt and stabilize training across diverse tasks. Yellow stars indicate ECHO's early high points, emphasizing that strong performance is achieved well before the full training budget is used. We note that for GSPO, even when using the clipping thresholds recommended in its original paper, we still observed sequence length collapse; therefore, its results are not displayed.

The superior performance and convergence efficiency of ECHO can be attributed to its carefully designed optimization mechanism. By leveraging the mini-batch ratio instead of token-level or sequence-level ratios, ECHO captures the overall likelihood shift between policies. This approach reduces local variance and provides a more faithful reflection of policy distribution changes, preventing noisy updates that could destabilize training. Compared to GSPO, which applies sequence-level importance sampling coefficients uniformly to each token, ECHO avoids the excessive pruning of learning signals, as it computes clipping based on batch-level statistics. As a result, ECHO maintains access to a larger fraction of informative samples, improving both stability and efficiency.

The batch-level clipping mechanism further stabilizes optimization by suppressing extreme updates while preserving tokens with relatively stable probabilities across the old and new policies. In GSPO, sequence-level clipping can still result in a large number of tokens being effectively ignored when facing unstable sequences, whereas ECHO's *clip-at-batch* approach ensures that individual token updates are constrained only when truly necessary. This retains informative low-probability tokens, maintaining access to subtle yet important signals and reducing variance accumulation across the batch.

Although clipping is conducted at the batch level, ECHO preserves token-level importance sampling. Each token's gradient is weighted according to its importance within the sequence rather than applying uniform treatment across the entire batch. This *learn-by-token* mechanism allows the model to distinguish between different tokens even when using coarse-grained batch-level constraints. Consequently, the model can capture fine-grained patterns, such as the role of numbers or

logical symbols in equations, and dynamically focus on high-entropy tokens that carry richer information. The gradual accumulation of these signals throughout training facilitates effective learning of complex sequential structures.

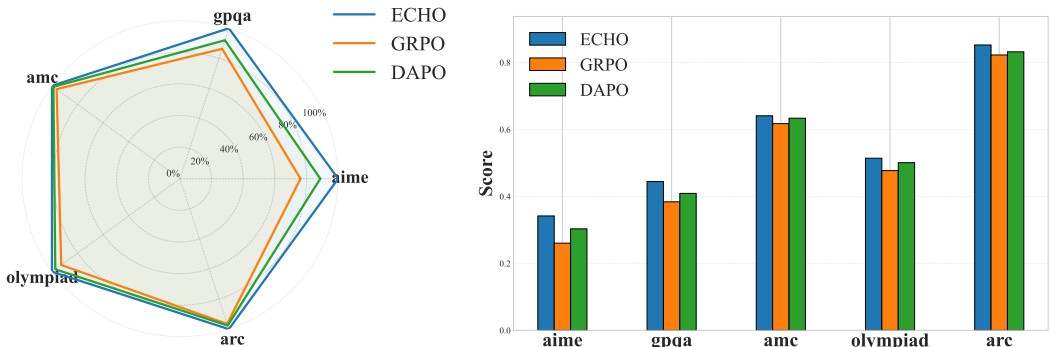

Figure 3: Performance of models across five benchmarks: AIME2024, AMC, GPQA, Olympiad, and ARC. (a) Radar chart showing per-benchmark results. (b) Bar chart comparing model performance across benchmarks. Overall, ECHO demonstrates competitive performance, surpassing both DAPO and GRPO on both Math and OOD datasets.

In addition, batch-level reweighting introduces a smoothing effect that implicitly normalizes updates, producing a more stable gradient distribution. By combining global stability with local flexibility, ECHO mitigates the variance accumulation issues that GSPO faces under sequence-level clipping, ensuring learning progresses efficiently while remaining sensitive to critical token-level features.

Together, these design choices establish a *clip-at-batch, learn-by-token* paradigm. This mechanism not only improves peak performance but also accelerates convergence, reduces the risk of losing informative tokens, and achieves more reliable optimization compared to prior methods, explaining the observed rapid and stable training dynamics.

## 6 CONCLUSION

We present ECHO, a coarse-grained hybrid optimization method designed to stabilize reinforcement learning for large language models. By leveraging batch-level clipping, token-level importance, and batch-level reweighting, ECHO effectively mitigates gradient noise and variance accumulation, enabling faster convergence and higher peak performance. Extensive experiments across multiple in-domain Math benchmarks and out-of-distribution reasoning datasets demonstrate that ECHO consistently outperforms prior RL-based and post-training methods, highlighting its effectiveness in both preserving strong in-distribution capabilities and improving generalization.

Beyond its immediate empirical benefits, ECHO offers promising potential for future applications in large-scale model training. Its *clip-at-batch, learn-by-token* paradigm can be naturally integrated into mixture-of-experts (MoE) training or other distributed RL frameworks, where stable gradient propagation and token-level signal preservation are critical. Furthermore, ECHO's ability to maintain high-entropy tokens and fine-grained sequence information suggests that it could facilitate more efficient adaptation in multi-task and continual learning scenarios, making it a versatile tool for advancing LLM optimization.

## ETHICS STATEMENT

This work does not involve personally identifiable information or sensitive user data. All datasets used are either publicly available or synthetically generated, and no private or proprietary data were accessed. We are aware of potential ethical concerns in large language model training, including fairness, bias, and misuse. To mitigate these risks, we carefully followed dataset usage licenses, conducted filtering to remove toxic content, and restricted evaluation to research purposes. We believe our contributions promote safe and responsible development of learning algorithms.

## REPRODUCIBILITY STATEMENT

We have made extensive efforts to ensure the reproducibility of our work. All hyperparameters, training details, and model architectures are fully described in the main text and appendix. We additionally provide pseudocode and mathematical derivations of the optimization objective. Experimental results are averaged across multiple seeds, and evaluation datasets are standard public benchmarks. Upon acceptance, we will release code, training scripts, and detailed instructions to enable full replication of our experiments.

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

# A APPENDIX

## A.1 LLM USAGE

To enhance the clarity and readability of this manuscript, we employed large language models (LLMs) for text refinement. Importantly, all technical content, experimental design, and results were generated, verified, and interpreted solely by the authors. The use of LLMs was strictly limited to language polishing and did not influence the scientific contributions of this work.

## A.2 GRADIENT DERIVATION IN THE UNCLIPPED REGION

We recall the unclipped ECHO objective:

$$\mathcal{J}_{\text{ECHO}}(\theta) = \mathbb{E}\big[C(\theta)\,L(\theta)\big], \quad C(\theta) = \frac{R_{\text{batch}}(\theta)}{R_{\text{batch}}(\theta)^{\dagger}}, \quad L(\theta) = \frac{1}{N}\sum_{i,t}\rho_{i,t}(\theta)\,\hat{A}_{i,t},$$

where

$$R_{\text{batch}}(\theta) = \frac{1}{N}\sum_{i,t}\rho_{i,t}(\theta), \quad \rho_{i,t}(\theta) = \frac{\pi_\theta(o_{i,t}\mid q, o_{i,<t})}{\pi_{\theta_{\text{old}}}(o_{i,t}\mid q, o_{i,<t})},$$

and $N = \sum_{i=1}^{G}|o_i|$ is the total number of tokens in the batch.

**Product rule.** Applying the product rule yields

$$\nabla_\theta \mathcal{J}_{\text{ECHO}}(\theta) = \mathbb{E}\big[\,C(\theta)\,\nabla_\theta L(\theta) + L(\theta)\,\nabla_\theta C(\theta)\,\big].$$

**First term: $\nabla_\theta L(\theta)$.** We compute the gradient of $L(\theta)$:

$$\nabla_\theta L(\theta) = \nabla_\theta\left(\frac{1}{N}\sum_{i,t}\rho_{i,t}(\theta)\,\hat{A}_{i,t}\right) \tag{16}$$

$$= \frac{1}{N}\sum_{i,t}\hat{A}_{i,t}\,\nabla_\theta\rho_{i,t}(\theta) \tag{17}$$

To compute $\nabla_\theta\rho_{i,t}(\theta)$, note that $\pi_{\theta_{\text{old}}}$ is constant with respect to $\theta$:

$$\nabla_\theta\rho_{i,t}(\theta) = \nabla_\theta\frac{\pi_\theta(o_{i,t}\mid q, o_{i,<t})}{\pi_{\theta_{\text{old}}}(o_{i,t}\mid q, o_{i,<t})} \tag{18}$$

$$= \frac{1}{\pi_{\theta_{\text{old}}}(o_{i,t}\mid q, o_{i,<t})}\nabla_\theta\pi_\theta(o_{i,t}\mid q, o_{i,<t}) \tag{19}$$

Now applying the log-derivative trick $\nabla_\theta\pi_\theta = \pi_\theta\nabla_\theta\log\pi_\theta$:

$$\nabla_\theta\rho_{i,t}(\theta) = \frac{1}{\pi_{\theta_{\text{old}}}(o_{i,t}\mid q, o_{i,<t})}\cdot\pi_\theta(o_{i,t}\mid q, o_{i,<t})\cdot\nabla_\theta\log\pi_\theta(o_{i,t}\mid q, o_{i,<t}) \tag{20}$$

$$= \frac{\pi_\theta(o_{i,t}\mid q, o_{i,<t})}{\pi_{\theta_{\text{old}}}(o_{i,t}\mid q, o_{i,<t})}\nabla_\theta\log\pi_\theta(o_{i,t}\mid q, o_{i,<t}) \tag{21}$$

$$= \rho_{i,t}(\theta)\nabla_\theta\log\pi_\theta(o_{i,t}\mid q, o_{i,<t}) \tag{22}$$

Therefore:

$$\nabla_\theta L(\theta) = \frac{1}{N}\sum_{i,t}\hat{A}_{i,t}\,\rho_{i,t}(\theta)\,\nabla_\theta\log\pi_\theta(o_{i,t}\mid q, o_{i,<t}) \tag{23}$$

**Second term: $\nabla_\theta C(\theta)$.** Since $R_{\text{batch}}(\theta)^{\dagger}$ is detached (stop-gradient), we have:

$$\nabla_\theta C(\theta) = \nabla_\theta\left(\frac{R_{\text{batch}}(\theta)}{R_{\text{batch}}(\theta)^{\dagger}}\right) \tag{24}$$

$$= \frac{1}{R_{\text{batch}}(\theta)^{\dagger}}\nabla_\theta R_{\text{batch}}(\theta) \tag{25}$$

$$= \frac{1}{R_{\text{batch}}(\theta)^{\dagger}}\nabla_\theta\left(\frac{1}{N}\sum_{j,k}\rho_{j,k}(\theta)\right) \tag{26}$$

$$= \frac{1}{R_{\text{batch}}(\theta)^{\dagger}}\frac{1}{N}\sum_{j,k}\nabla_\theta\rho_{j,k}(\theta) \tag{27}$$

Using the same result as above for $\nabla_\theta \rho_{j,k}(\theta)$:

$$\nabla_\theta C(\theta) = \frac{1}{R_{\text{batch}}(\theta)^\dagger} \frac{1}{N} \sum_{j,k} \rho_{j,k}(\theta) \nabla_\theta \log \pi_\theta(o_{j,k} \mid q, o_{j,<k}) \tag{28}$$

Note that $\frac{1}{N} \sum_{j,k} \rho_{j,k}(\theta) = R_{\text{batch}}(\theta)$, so:

$$\nabla_\theta C(\theta) = \frac{1}{R_{\text{batch}}(\theta)^\dagger} \cdot R_{\text{batch}}(\theta) \cdot \frac{1}{N} \sum_{j,k} \nabla_\theta \log \pi_\theta(o_{j,k} \mid q, o_{j,<k}) \tag{29}$$

$$= \frac{R_{\text{batch}}(\theta)}{R_{\text{batch}}(\theta)^\dagger} \cdot \frac{1}{N} \sum_{j,k} \nabla_\theta \log \pi_\theta(o_{j,k} \mid q, o_{j,<k}) \tag{30}$$

$$= C(\theta) \cdot \frac{1}{N} \sum_{j,k} \nabla_\theta \log \pi_\theta(o_{j,k} \mid q, o_{j,<k}) \tag{31}$$

**Combined gradient.** Substituting both components gives:

$$\nabla_\theta \mathcal{J}_{\text{ECHO}}(\theta) = \mathbb{E}\Bigg[ C(\theta) \frac{1}{N} \sum_{i,t} \hat{A}_{i,t} \rho_{i,t}(\theta) \nabla_\theta \log \pi_\theta(o_{i,t} \mid q, o_{i,<t}) \tag{32}$$

$$+ L(\theta) C(\theta) \frac{1}{N} \sum_{j,k} \nabla_\theta \log \pi_\theta(o_{j,k} \mid q, o_{j,<k}) \Bigg] \tag{33}$$

Since the summation indices $(i,t)$ and $(j,k)$ both run over all tokens in the batch, we can combine them:

$$\nabla_\theta \mathcal{J}_{\text{ECHO}}(\theta) = \mathbb{E}\Bigg[ C(\theta) \frac{1}{N} \sum_{i,t} \Big( \hat{A}_{i,t} \rho_{i,t}(\theta) + L(\theta) \Big) \nabla_\theta \log \pi_\theta(o_{i,t} \mid q, o_{i,<t}) \Bigg] \tag{34}$$

**Alternative compact forms.** We can also express this as:

$$\nabla_\theta \mathcal{J}_{\text{ECHO}}(\theta) = \mathbb{E}\Bigg[ \frac{1}{N R_{\text{batch}}(\theta)^\dagger} \sum_{i,t} \rho_{i,t}(\theta) \big( R_{\text{batch}}(\theta) \hat{A}_{i,t} + L(\theta) \big) \nabla_\theta \log \pi_\theta(o_{i,t} \mid q, o_{i,<t}) \Bigg] \tag{35}$$

Or equivalently:

$$\nabla_\theta \mathcal{J}_{\text{ECHO}}(\theta) = \mathbb{E}\Bigg[ C(\theta) \frac{1}{N} \sum_{i,t} \rho_{i,t}(\theta) \Big( \hat{A}_{i,t} + \frac{L(\theta)}{R_{\text{batch}}(\theta)} \Big) \nabla_\theta \log \pi_\theta(o_{i,t} \mid q, o_{i,<t}) \Bigg] \tag{36}$$

**Interpretation.** Each token $(i,t)$ contributes to the gradient with weight:

$$w_{i,t}(\theta) = C(\theta) \rho_{i,t}(\theta) \Big( \hat{A}_{i,t} + \frac{L(\theta)}{R_{\text{batch}}(\theta)} \Big)$$

This weight structure provides:

- **Batch-level modulation:** $C(\theta)$ adaptively rescales updates according to the global policy shift.
- **Token-level importance:** Each token contributes proportionally to its importance ratio $\rho_{i,t}(\theta)$ and advantage $\hat{A}_{i,t}$.
- **Variance reduction:** The term $L(\theta)/R_{\text{batch}}(\theta)$ acts as a batch-level baseline that reduces gradient variance.
- **Stability control:** When $R_{\text{batch}}(\theta)$ grows large, $C(\theta)$ decreases in the clipped case, preventing overly aggressive updates.

