# OpenReview forum: "ECHO: Efficient Coarse-Grained Hybrid Optimization — Clip at Batch, Learn at Token"
_ICLR.cc/2026/Conference — ICLR 2026 Conference Withdrawn Submission_

### Official Review · Reviewer_AUgi · 2025-10-30

**Soundness:** 3
**Presentation:** 3
**Contribution:** 2
**Rating:** 4
**Confidence:** 4

**Summary:**

The paper proposes ECHO (Efficient Coarse-Grained Hybrid Optimization), a new reinforcement learning framework for large language model (LLM) alignment. The core idea is to apply batch-level clipping of importance sampling ratios, stabilizing global policy updates, while preserving token-level importance weighting to maintain fine-grained learning signals.
.
- The paper derives the gradient formulation of ECHO, showing that batch-level correction ensures stability while token-wise sampling enables precise credit assignment.
- Experiments on nine benchmarks (AIME’24/25, AMC’23, MATH, OlympiadBench, Minerva, ARC-C, GPQA, MMLU-Pro) demonstrate that ECHO achieves faster convergence, higher stability, and better generalization than GRPO, GSPO, and DAPO.

**Strengths:**

**Novel hybrid optimization design:** Integrates batch-level clipping with token-wise learning, balancing stability and expressiveness.
- **Improved training stability:** Mitigates gradient explosion and avoids sequence length collapse observed in GSPO or GRPO.
- **Higher sample efficiency:** Reaches strong performance with significantly fewer steps compared to DAPO.
- **Scalable and practical:** Compatible with large-scale distributed RL training and adaptable to future MoE or multi-task setups.

**Weaknesses:**

- The experimental section is relatively weak. Since the paper focuses on the effect of clipping, additional ablation studies would strengthen the analysis — for example, showing the clipping ratio dynamics or policy loss curves over training.

- Minor weakness: The paper emphasizes training stability, yet Figure 2 appears to show results from a single random run. It would be more convincing to include mean curves with variance/error bars across multiple seeds.

**Questions:**

- Could the authors provide the clipping ratios used for different algorithms? I believe this information is important for analyzing and comparing the optimization behavior.
- Why does ECHO achieve faster convergence in the early training stage? In my understanding, at the beginning of training, the clipping ratios across different algorithms should be close to 0 (i.e., little divergence between policies). Therefore, I would expect the acceleration effect to appear in the mid-stage, when the clipping ratios start to diverge.

---

### Official Review · Reviewer_TJSF · 2025-11-01

**Soundness:** 3
**Presentation:** 3
**Contribution:** 2
**Rating:** 4
**Confidence:** 4

**Summary:**

The paper effectively identifies a key limitation in existing methods: GRPO's token-level clipping and GSPO's uniform sequence-level treatment. The proposed hybrid approach, ECHO, is a promising direction. However, the articulation of its core novelty—the "batch-level clipping" mechanism—requires further precision.

**Strengths:**

1. The experimental results are comprehensive, spanning multiple benchmarks, and demonstrate strong performance. The critique that GSPO suffers from "severe sequence length collapse" is valid and a meaningful practical drawback.

2. The gradient derivation in Section 3.3 is technically sound, and the interpretation of "coarse-to-fine" control is insightful. That said, the paper would benefit from a more thorough discussion of computational overhead and implementation complexity.

**Weaknesses:**

1. The derivation indicates that in the unclipped regime, the batch-level correction factor approaches. This suggests that the primary effect of ECHO is gradient stabilization and variance reduction via the introduced baseline $\frac{L(\theta)}{R_{\text{batch}}(\theta)}$, rather than a fundamentally new clipping paradigm. To strengthen the paper, the authors should explicitly distinguish their theoretical contribution.

2. By excluding GSPO from several comparisons (e.g., Table 1), the evaluation sidesteps a direct comparison against a properly stabilized version of this baseline. To present a more compelling case for ECHO's superiority, the authors should include a comparison with a stabilized GSPO variant. This could involve techniques like length normalization or entropy regularization. Such a comparison is necessary to ensure that ECHO's performance gains are attributable to its hybrid design, and not merely to the avoidance of a known, but correctable, failure mode in a specific baseline implementation.

3. Calculating the batch-level average log-probability ratio $R_{\text{batch}}(\theta)$ requires aggregating statistics across all tokens in the entire mini-batch. In a distributed training setup, this could introduce non-trivial synchronization and communication costs. The authors should analyze or discuss ECHO's practical training efficiency—specifically, its steps/second and memory footprint relative to GRPO and DAPO. This would assure readers that the improved convergence and performance are not achieved at a prohibitive computational cost.

**Questions:**

1. To strengthen the paper, the authors should more explicitly distinguish their theoretical contribution. Is it primarily a clever synthesis and re-weighting of existing ideas (i.e., combining sequence-level statistics with token-level gradients), or does it introduce a genuinely new optimization operator? A clearer framing would help the reader accurately assess the conceptual advance.

---

### Official Review · Reviewer_TW5Y · 2025-11-03

**Soundness:** 3
**Presentation:** 2
**Contribution:** 2
**Rating:** 4
**Confidence:** 4

**Summary:**

This paper proposes a variation of GRPO that combines token-level importance sampling and batch-level clipping. The authors claim that this design leads faster convergence and more stable training.

**Strengths:**

* Experiments demonstrate notable improvements in training efficiency, particularly during the early stages of training.

**Weaknesses:**

* The main equations in section 3.2 are not clearly presented with some notation left undefined. Moreover, according to line 204-205, the authors first apply batch-level clipping to the token-level importance ratio, and then perform another token-level clipping on top. This implementation appears inconsistent with the method described in the paper. These equations make it difficult to follow the proposed method.
* The authors claim the proposed method is superior because 1/ it preserves more learning signal comparing to token-level clipping, leading to faster convergence, and 2/ it leads to more stable training. However, the paper does not provide sufficient justification or theoretical explanation for these claims, and the two advantages can appear contradictory—preserving more learning signal can increase variance, which can reduce stability.

**Questions:**

Experiments are conducted on a 7B model; however, RL training instability often becomes more pronounced at larger model scales. Since the proposed method relaxes the clipping mechanism, is it a valid concern that it may increase the risk of instability when applied to larger models?

---

### Official Review · Reviewer_8syB · 2025-11-03

**Soundness:** 2
**Presentation:** 2
**Contribution:** 2
**Rating:** 2
**Confidence:** 3

**Summary:**

This paper proposes to augment the token-wise importance ratio with a batch-wise importance ratio for LLM alignment. The empirical evaluation on mathematics and logic reasoning datasets demonstrates the competitive performance of the proposed method.

**Strengths:**

- Considering batch-wise information during LLM alignment is novel

- The empirical evaluation on two Qwen models and 9 datasets demonstrates the competitive performance of the proposed method

**Weaknesses:**

- I appreciate that the authors provide Figure 1 to motivate the method. However, the results in Figure 1 do not well support the corresponding claim (see my questions below).

- The writing about the method is redundant with repeated content:
1) In Section 3, some notations are defined first, being used in the later equations, but are expanded again. For example, Eq. (6) defines $R_{batch}(\theta)$. Then, $R_{batch}(\theta)$ is used in Eq. (8), but is expanded in Eq. (9). The intention of doing this is not explained. This presentation is implicitly redundant.
2) The ECHO objective function is introduced in 204 and is fully presented again in line 214. The gradient of the ECHO objective function is derived in line 227 and fully presented again in line 241.

- The experiment mainly presents the benchmark scores. Many claims in the text (e.g., Section 6, "highlighting its effectiveness in both preserving strong in-distribution capabilities and improving generalization") need more evidence to support.

**Questions:**

- In the caption of Figure 1, the author writes "(a)..., showing that GRPO ... while GSPO and ECHO ...". Figure 1 (a) has only one line for GRPO. How does this single line provide a comparison to the other two methods? Also in the corresponding text (Section 1, paragraph 2): "token-level... in an excessive proportion of tokens being truncated". How large is the proportion, and is it really "excessive"?


# Minors
- "Eq. equation" is frequently used in the paper. Use either "Eq." or "Equation"
- Section 1, paragraph 3, line 1. "im po a r tan ce"
- page 3, line 117, "baseline" is not defined.
- page 4, line 175. The definition of $\tilde{\rho}_{i, t} (\theta)$ is described in Eq. 8-10, instead of Eq. 10.
- page 4, line 192. The superscript of the denominator in Eq. 8 is not defined.
- page 5, line 232. "$\rho_{i, t}$" should be $\rho_{i, t}(\theta)$.
- page 5, line 252, "under normal circumstances", consider avoid the word "normal" as it is ambiguous.

---

### Note · Authors · 2025-11-17

I have read and agree with the venue's withdrawal policy on behalf of myself and my co-authors.